# Functionalized GO Membranes for Efficient Separation of Acid Gases from Natural Gas: A Computational Mechanistic Understanding

**DOI:** 10.3390/membranes12111155

**Published:** 2022-11-16

**Authors:** Quan Liu, Zhonglian Yang, Gongping Liu, Longlong Sun, Rong Xu, Jing Zhong

**Affiliations:** 1Analytical and Testing Center, School of Chemical Engineering, Anhui University of Science and Technology, Huainan 232001, China; 2State Key Laboratory of Materials-Oriented Chemical Engineering, College of Chemical Engineering, Nanjing Tech University, 30 Puzhu Road (S), Nanjing 211816, China; 3Key Laboratory of Advanced Catalytic Materials and Technology, School of Petrochemical Engineering, Changzhou University, Gehu Road, Changzhou 213164, China

**Keywords:** acid gas removal, graphene oxide, membrane separation, molecular simulation, natural gas

## Abstract

Membrane separation technology is applied in natural gas processing, while a high-performance membrane is highly in demand. This paper considers the bright future of functionalized graphene oxide (GO) membranes in acid gas removal from natural gas. By molecular simulations, the adsorption and diffusion behaviors of several unary gases (N_2_, CH_4_, CO_2_, H_2_S, and SO_2_) are explored in the 1,4-phenylenediamine-2-sulfonate (PDASA)-doped GO channels. Molecular insights show that the multilayer adsorption of acid gases evaluates well by the Redlich-Peterson model. A tiny amount of PDASA promotes the solubility coefficient of CO_2_ and H_2_S, respectively, up to 4.5 and 5.3 mmol·g^−1^·kPa^−1^, nearly 2.5 times higher than those of a pure GO membrane, which is due to the improved binding affinity, great isosteric heat, and hydrogen bonds, while N_2_ and CH_4_ only show single-layer adsorption with solubility coefficients lower than 0.002 mmol·g^−1^·kPa^−1^, and their weak adsorption is insusceptible to PDASA. Although acid gas diffusivity in GO channels is inhibited below 20 × 10^−6^ cm^2^·s^−1^ by PDASA, the solubility coefficient of acid gases is certainly high enough to ensure their separation efficiency. As a result, the permeabilities (*P*) of acid gases and their selectivities (*α*) over CH_4_ are simultaneously improved (*P*_CO2_ = 7265.5 Barrer, *α_CO2/CH4_ =* 95.7; *P_(_*_H2S+CO2)_ = 42075.1 Barrer, *α_H2S/CH4_* = 243.8), which outperforms most of the ever-reported membranes. This theoretical study gives a mechanistic understanding of acid gas separation and provides a unique design strategy to develop high-performance GO membranes toward efficient natural gas processing.

## 1. Introduction

Methane (CH_4_), as the main constituent of natural gas, is one kind of renewable energy source [1]. The raw natural gas coming from crude oil wells always exists in the form of mixtures, containing other light hydrocarbons, nitrogen (N_2_), carbon dioxide (CO_2_), hydrogen sulfide (H_2_S), and sulfur dioxide (SO_2_). Among these impurities, significant amounts of CO_2_, H_2_S, and SO_2_ commonly called acid gases are the most harmful components in raw natural gas, which not only lowers the calorific value of CH_4_ but also causes internal corrosion in gas pipelines [2,3]. Therefore, to meet the requirements of end users and the specifications of transportation pipelines, the removal of acid gases is an essential process in natural gas processing [3,4]. Several processes can be adopted to remove acid gases, including pressure swing adsorption, supersonic separation, and membrane separation. In addition, natural gas can also be purified by forming CO_2_ hydrates from the gas mixtures [5,6,7]. The commercialized technology is amine scrubbing [8], which uses plenty of alkanolamine solutions in absorption columns to dissolve acid gases. However, it requires the use of large equipment, rapidly increasing the operating cost [9], and lots of undesirable liquid wastes produced in this process pose a threat to the environment. Alternatively, with low energy consumption, low pollution and high separation efficiency, membrane gas separation technology is regarded as a potential candidate for acid gas removal [10]. Especially under ordinary operation conditions (i.e., room temperature and low operating pressure), it will achieve better economic benefits in natural gas processing.

Various membrane materials have been developed to address these challenging separations, such as polymer [11,12], metal-organic framework (MOF) [13,14] and graphene [15]. Among them, the polymeric membrane is the most large-scale development for commercial, while its performance is somewhat low primarily due to the trade-off effect. Fortunately, two-dimensional (2D) graphene oxide (GO) membranes with tailorable channels and abundant active sites are emerging candidates for boosting molecular separation performance [15,16]. It is reported that their inherent transport channels can be regulated for selective permeation at the sub-nanometer scale [17]. For instance, by adjusting ultraviolet irradiation, the interlayer spacing of GO membrane was precisely controlled by Zheng et al. to improve the separation efficiency of these two species with a very low molecular weight difference [18]. Our previous work also showed that the 1,4-phenylenediamine-2-sulfonate (PDASA)-functionalized GO channels facilitated the adsorption of the polar molecule (i.e., water), and then largely promoted its permeation [19]. For acid gas removal, the CO_2_ permeability was successfully enhanced by incorporating GO nanosheets as the filler to create additional gas transport channels in polymers of intrinsic microporosity [20]. Additionally, using the strong affinity between GO and CO_2_ was a brilliant strategy to enhance the CO_2_ solubility in polyimide hybrid membranes [21]. After doping GO nanosheets, the CO_2_/CH_4_ separation performance of various polymeric membranes was promoted to outperform the 2008 Robeson upper bound [15,22].

However, as mentioned above, the GO nanosheet is mostly dispersed as a filler into mixed matrix membranes or prepared as hybrid membranes to separate CO_2_/CH_4_ [15,20,21,23], thus lack of exploration on pure GO membrane especially on its separation mechanism for acid gas removal. Fortunately, a few molecular simulations attempted to explore the CO_2_/CH_4_ separation process through pure GO membranes [24,25]. Whereas, for other 2D membranes, most previous simulations demonstrated that there were two main dominated separation mechanisms (i.e., the size-sieving effect and preferential adsorption) in natural gas processing [26,27,28]. A suitable aperture is key to the high separation performance of CO_2_/CH_4_ [26,27]. While in order to further improve the removal efficiency of CO_2_, the separation mechanism should be governed by preferential adsorption, which helps to improve CO_2_ separation selectivity [28]. However, until now, there has been no theoretical model established for acid gas separation through GO membranes. Therefore, in order to establish this theoretical model, it is necessary to study the acid gas permeation behavior in GO channels from the perspectives of adsorption and diffusion. Moreover, CO_2_ and other acid gases (i.e., H_2_S and SO_2_) need to be studied at the same time. Furthermore, to improve the removal efficiency, a rational design of a GO membrane at the molecular level is highly in demand. This study aims to theoretically design a high-performance GO membrane toward acid gas removal and explore the separation models.

In this work, GO membranes are functionalized by PDASA (this selection is inspired by our previous experimental work [19]) to examine how it performs in removing acid gases (CO_2_, H_2_S, and SO_2_) from CH_4_ and N_2_. By Grand Canonical Monte Carlo (GCMC) simulations, unary isotherms of different gases in GO membranes with variable doping amounts of PDASA are first studied by several adsorption models. To accurately describe the adsorption characteristics of different gases and provide molecular insights, structural and energetic analyses are conducted in GO channels via molecular distribution probability, radial distribution function (RDF), isosteric heat, and hydrogen bonds. The solubility coefficient is calculated to characterize the adsorption ability of different gases. Then gas diffusion behavior is explored by molecular dynamical (MD) simulations. After that, the acid gas separation performance is predicated on the basis of the solution-diffusion mechanism. Finally, a performance comparison with previous reports is enclosed to demonstrate the potential of the PDASA-doped GO membranes in natural gas processing.

## 2. Models and Methods

Figure 1 shows the simulation models. First of all, GO nanosheets with the format of C_312_(O)_65_(OH)_79_(COOH)_4_ were constructed by the Material studio in amorphous cell as per our previous works [16,29,30,31,32,33]. Functional groups were randomly distributed on the sp^2^-conjugated surface of which the dimensions were 3 × 3 nm^2^, as shown in Figure 1a. The numbers of epoxy, hydroxyl and carboxyl groups were 65, 79, and 4, respectively, similar to our previous experimental reports [19]. As a result, the oxidized ratio that was defined by the total number of oxygen atoms to carbon atoms was about 0.48, which is feasible in membrane process simulation for both gas and liquid separations [16,32,33]. Five gases with variable electronegativities and kinetic diameters were investigated, as shown in Figure 1b. Electrostatic potentials show that the acid gases of CO_2_, H_2_S and SO_2_ exhibit higher electronegativity compared to CH_4_ and N_2_. To reveal gas sorption and diffusion behaviors in the lamellar structure of GO membranes, two GO nanosheets were parallelly aligned with interlayer spacing initially set as 0.8 nm (Figure 1c). To increase the affinity between GO membrane and acid gases, interlayer channel was functionalized with PDASA groups (Figure 1f) that have a great affinity to polar molecules [19]. The number of doped PDASA molecules increased from 1 to 5, correspondingly to the doping amounts varying from 1.5 to 7.5 wt%. The atomic positions of GO nanosheets were flexible during simulations. After being loaded with PDASA groups, GO membranes were relaxed well, and then interlayer spacing was slightly enlarged, as shown in Figure 1d,e where the doping amounts are 4.5 wt% and 7.5 wt%, respectively.

Before GCMC simulations, GO membranes and gases were performed with geometry optimization to search for a minimum energy structure. In this process, the convergence thresholds of energy, force and displacement were specified as 10^−5^ kcal/mol, 10^−3^ kcal/mol/Å and 10^−5^ Å, respectively. To calculate adsorption isotherms of gases in flexible GO membranes, the Configurational bias method [34] was performed with 10^7^ equilibration and production steps. The temperature was maintained at 298 K by the algorithm of Nosé-Hoover thermostat [35]. Production frame was output every 10,000 steps. Partial charges were taken from the Compass force field [36], which was also used to describe interatomic interactions among membrane and variable gases. Here, nonbonded interactions were summarized by electrostatic and van der Waals potentials. Long-range electrostatic interactions were handled with the Ewald method [37] with an accuracy of 10^−5^ kcal/mol, whereas van der Waals interaction potentials were predicated by the atom-based method with a 9.8 Å cut-off distance. Periodic boundary conditions are applied in all three directions. After adsorption simulations, the lowest energy configuration returned from the GCMC calculation was used as the initial frame to explore gas diffusion properties. In MD simulations, there were a total of 50 gas molecules inserted in GO membranes and they could freely roam in GO interlayers. The system reached temperature (298 K) equilibrium first in an isothermal-isobaric ensemble for 1 ns. The pressure was controlled at 1 bar by the Berendsen barostat [38] with a decay constant of 0.1 ps. Subsequently, the production runs were performed in a canonical ensemble. The time step was set as 0.5 fs and trajectories were recorded every 2 ps, and the total simulation time was 2 ns. The final results were averaged over three independent trials.

## 3. Results and Discussion

### 3.1. Adsorption Evaluation

To calculate the adsorption isotherms of different gases in GO membranes, GCMC simulations were performed under low pressures (0.01 KPa~1000 Kpa). The fugacity coefficients of unary gases (N_2_, CH_4_, CO_2_, H_2_S and SO_2_) are close to 1.0 under these pressures by physical property estimation in Aspen using the Peng-Robinson equation-of-state [39], indicating that the gas behavior approximates the ideal gas model. Therefore, the fugacity and pressure are approximately equal. Figure 2 shows the absolute adsorption isotherms of five gases are dependent on the relative pressures in GO membranes with variable doping amounts of PDASA. The adsorption capacities of CH_4_ and N_2_ slowly rise with increasing pressure. While for acid gases (CO_2_, H_2_S and SO_2_), their isotherms grow rapidly, especially a sudden increase at relatively low pressures, behaving in a different adsorption mode. As a result, the adsorption capacities of acid gases in GO membranes are obviously larger than those of CH_4_ and N_2_. In addition, the maximum absorption capacity increases in the order of N_2_ < CH_4_ < CO_2_ < SO_2_ < H_2_S. With increasing the doping amounts of PDASA from 0.0 to 7.5 wt%, the adsorption capacities of three acid gases increase at first and then decrease, as shown in Figure 1a–f. In view of the low density of adsorbed gases at low pressure and low temperature, the absolute adsorption capacity (*Q_ab_*) obtained in our simulations is close to the excess adsorption capacity (*Q_ex_*) that is determined in the experiment according to Equation (1) [40] where *ρ_g_* is the gas density at simulated pressure and *V_f_* is the free volume in GO membranes. Therefore, the absolute adsorption isotherms in Figure 1 without further conversion can be directly described by adsorption models.
(1)$$Q_{e}{}_{x}\;=\;Q_{a}{}_{b}-\rho_{g}V_{f}$$
(2)$$S_{0}\;=\;\underset{p\to 0}{\lim}\frac{Q_{e}}{P}$$
$$Q_{e}{}_{x}\;=\;\delta P+\frac{\beta P}{1+\gamma P^{n}}\;=\;\left\{\begin{array}{l} \frac{Q_{L}K_{L}P}{1+K_{L}P}\;\;\;\delta \;=\;0;\;n\;=\;1\;(Langmuir,\;for\;CH_{4}\;and\;N_{2})\;\;\;\;\;\;\;\;\;\;(3)\\ \frac{\beta P}{1+\gamma P^{n}}\;\;\;\delta \;=\;0;\;0<n<1\;(Redlich-Peterson,\;for\;H_{2}S\;and\;CO_{2})\;\;\;\;\;(4)\\ \delta P+\frac{\beta P}{1+\gamma P^{n}}\;\delta \neq 0;\;0<n<1\;(Dual-mode,\;for\;SO_{2})\;\;\;\;\;\;\;\;\;\;\;\;\;(5) \end{array}\right\}$$
$$S_{0}\;=\;\left\{\begin{array}{l} Q_{L}K_{L}\;\;\;\;(for\;CH_{4}\;and\;N_{2})\;\;\;\;\;(6)\\ \beta \;\;\;\;\;\;(for\;H_{2}S\;and\;CO_{2})\;\;\;\;\;\;(7)\\ \delta +\beta \;\;\;\;\;\;\;(for\;SO_{2})\;\;\;\;\;\;\;\;(8) \end{array}\right\}$$

The solubility coefficient (S_0_) of infinite dilution is an important factor in characterizing membrane separation properties, which is defined as the slope of isotherm at infinite dilution (Equation (2)) [41,42,43]. When gas concentration is extremely low, several theoretical models (Equations (3)–(5)) are applied to fit isotherms to obtain the S_0_ of gases in GO membranes, where *P* is the sorbate pressure, and *δ*, *β* and *γ* are fitting parameters. After curve fitting, it shows that the adsorption of CH_4_ and N_2_ obey the Langmuir model [44] (Equation (3)) where *Q_L_* is the maximal adsorption capacity and *K_L_* is the adsorption equilibrium constant, indicating a simple adsorption process. While simulation results suggest a three-parameter model (i.e., Redlich-Peterson [45], Equation (4)) for CO_2_ and H_2_S, where *n* is the empirical constant. The adsorption behavior for SO_2_ is a little complex as it needs more variables to fit the isotherm based on the dual-mode sorption model [46] as Equation (5). All fitting parameters are presented in Appendix A. A high correlation coefficient (R^2^) above 0.992 for most systems indicates the reliability of these adopted adsorption models [45]. These different theoretical models are ascribed to the variable adsorption mechanism of gases in GO membranes, which will be discussed below. Thereafter, the S_0_ of different gases in GO membranes is accordingly calculated by Equations (6)–(8) [41,42,43].

### 3.2. Adsorption Insight

To quantitatively evaluate the adsorption ability of different gases in GO membranes and understand the variable adsorption models, Figure 3 presents the calculated S_0_ and the corresponding adsorption behaviors. The S_0_ as a function of variable doping amounts of PDASA is shown in Figure 3a. For CH_4_ and N_2_, the S_0_ values in different GO membranes are less than 0.002 mmol·g^−1^·kPa^−1^, almost invariable with the doped PDASA. The distribution probability in Figure 3b reveals that the particles of CH_4_ and N_2_ are highly concentrated, forming single-layer adsorption. Snapshots in Figure 3c,d provide a visual perspective for these single-adsorbate cases, where CH_4_ and N_2_ deposit in the center of GO channels, indicating a weak adsorption ability. That is the reason their adsorption behaviors in GO membranes can be accurately represented by Langmuir model [44]. On the contrary, CO_2_ and H_2_S exhibit a strong adsorption ability with the S_0_ all above 3.4 mmol·g^−1^·kPa^−1^. As seen in Figure 3a, when the PDASA-doping amount is 3.0 wt%, CO_2_ and H_2_S exhibit the maximum S_0_ values of 4.5 and 5.3 mmol·g^−1^·kPa^−1^, respectively, almost 2.5 times higher than those values of GO membranes without doping PDASA. Continuously increasing the doping amounts, the S_0_ shows a downward trend. The adsorption ability of SO_2_ in GO membranes is extremely strong as there is an almost vertical ascent motion at the start point of isotherms (Figure 2). Therefore, the S_0_ of SO_2_ are all above 80 mmol·g^−1^·kPa^−1^ and not compared in Figure 3a. Compared to CH_4_ and N_2_, for acid gases, their maximum distribution probability is not in the center of channels but on either side of the center. By visual of Figure 3e–g, CO_2_, H_2_S and SO_2_ present multilayer adsorption in GO channels. In addition, they also have a probability to distribute “outside” channels due to periodic boundary conditions. The above complex adsorption behavior of CO_2_ and H_2_S indicates a strong adsorption ability, thus deserving the Redlich-Peterson model [45,47].

To reveal the positive effect of PDASA on acid gas adsorption in GO membranes, RDF, isosteric heat and hydrogen bonds are analyzed in Figure 4 to provide molecular insight into the adsorption process. The dynamic binding process between gases and PDASA is evaluated with RDF graph g(r) based on Equation (9) [33], where *r* is the distance from species *i* to *j*, *N_i_* represents the number of species *i*, *N_ij_*(*r, r + Δr)* is the number of *i* around *j* within a shell and *V* is the volume. The RDF value is a measure of binding affinity, whereas a high RDF value means a strong affinity of PDASA to gases. As seen in Figure 4a, the affinity increases following the sequence of N_2_ ≈ CH_4_ < CO_2_ < H_2_S ≈ SO_2_. The high affinity of PDASA to acid gases is the primary reason for its positive effect on acid gas adsorption, while the weak guest-membrane affinities lead to the weak adsorption of CH_4_ and N_2_ in GO channels. Isosteric heat, a decisive factor of adsorption strength, is analyzed in Figure 4b. Obviously, the isosteric heats of five gases in GO membranes increase in the order of N_2_ < CH_4_ < CO_2_ ≈ H_2_S < SO_2_, confirming the strong adsorption strength of acid gases in GO membranes, especially for SO_2_. Besides the binding affinity and isosteric heat, the strong adsorption of acid gases is also related to hydrogen bonds. Based on these two geometrical criteria [16], (1) r(H⋅⋅⋅O) ≤ 0.35 nm; (2) α(O-H⋅⋅⋅O) ≤ 30°, hydrogen bonds in acid gases adsorption process are pictured in Figure 4c–e. A great number of hydrogen bonds are formed between GO membranes and acid gases. In addition, the doped PDASA also contributes to the formation of hydrogen bonds, as shown in Figure 4f, which further helps GO membranes to capture H_2_S. The above effects synergistically promote acid gas adsorption, while large doping amounts will decrease the effective adsorption sites and reduce the packing efficiency of acid gases in GO channels due to the narrowing of the passage, which will be discussed below.
(9)$$g_{i}{}_{j}(r)\;=\;\frac{N_{i}{}_{j}(r,r+\Delta r)V}{4\pi r^{2}\Delta rN_{i}N_{j}}$$

### 3.3. Diffusion Evaluation

Dynamical properties of gases in GO channels are evaluated by mean square displacement (MSD) according to Equation (10) [32,33] in which the N refers to the total number of particles and *r*_*i*_(*t*) − *r*_*i*_(*t*_0_) is the displacement distance of particle i from the initial state t_0_ to the final state t. As shown in Figure 5, the gas mobility in GO channels with variable doping amounts of PDASA follows the sequence of N_2_ ≈ CH_4_ > H_2_S > CO_2_ > SO_2_, which means the diffusion process is not governed by the size-sieving effect. The large mobilities of CH_4_ and N_2_ in GO channels are attributed to their weak interactions with GO membranes, thus resulting in low mass-transfer resistance. Although with smaller molecular size, acid gases exhibit slow mobility in that the strong interactions generate a large transport resistance [16]. After doping the PDASA into GO channels, the mobilities of all gases slow down. Diffusion coefficient (D) is another key role in determining separation performance, which is calculated by the linear slope of MSD based on Equation (11) [32,33]. Taking the cases in pure GO membrane as examples, the logarithmic form shown in Appendix A can be fitted linearly from 100 to 1000 ps with slopes larger than 0.94, indicating that the gas diffusion tends to stabilize and approach to a normal diffusion state [48]. Then the D can be obtained from this region in MSD curves. To uncover the diffusion mechanism of gases in GO channels, the quantitative diffusivity, accessible free volume (AFV) [49] and effective transport channels are analyzed in Figure 6. Figure 6a illustrates that the diffusion coefficient generally shows a decreasing trend with the increase in the PDASA-doping amount. For N_2_ and CH_4_, both have diffusion coefficients larger than 240 × 10^−7^ cm^2^·s^−1^ due to the low transfer resistance, which agrees well with previous work [25], demonstrating the reliability of our calculations. In contrast, for acid gases, their diffusivities in GO channels are relatively low. Especially for SO_2_, its dynamic motion is severely restricted with diffusion coefficients lower than 80 × 10^−7^ cm^−2^·s^−1^. The AFV in variable GO membranes as a function of probe radius is shown in Figure 6b based on Equation (12) where V_f_ and V_o_ denote the free and occupied volumes, respectively. It shows that the AFV is sensitive to the probe radius. In addition, when the probe radius is larger than the molecular sizes of acid gases, the AFV nearly declines with the increase in the PDASA-doping amounts (Figure 6c). Figure 6d–i show the visualization of free volume. Apparently, the PDASA severed as barriers in GO channels to block the passage of gases (green region). With increasing the doping amounts, the effective passage is narrowed especially in GO-7.5 wt% PDASA (Figure 6i). That is the reason molecular diffusion is severely inhibited by doping PDASA in GO channels. This confirms that doping PDASA into GO channels brings a change not only in their adsorption but also in their diffusion. However, in this condition, diffusion is not supposed to govern the separation process of acid gases through the PDASA-doped GO membranes.
(10)$$MSD(t)\;=\;\frac{1}{N}\langle \overset{N}{\underset{i\;=\;1}{\sum }}{[r_{i}(t)-r_{i}(t_{0})]}^{2}\rangle$$
(11)$$D_{}\;=\;\frac{1}{6}\underset{t\to \infty }{\lim}\frac{dMSD}{dt}$$
(12)$$AFV\;=\;\frac{V_{f}}{V_{f}+V_{o}}\times 100\%\;$$

### 3.4. Separation Performance Prediction

The permeability coefficient, *P_i_*, with a typically reported unit of Barrer is determined on the basis of the solution-diffusion model in Equation (13), where the corresponding *S_i_* and *D_i_* have a unit of cm^3^(STP)·cm^−3^·mmHg and 10^−7^ cm^2^·s^−1^, respectively, which are included in Appendix A. The ideal gas selectivity, *α_i/j_*, is defined as the ratio of permeabilities of *i* and *j* by Equation (14). The separation performance of acid gases (CO_2_ and H_2_S) through PDASA-doped GO membranes is predicated in Figure 7. For CH_4_ and N_2_, their permeabilities are relatively low, as shown in Figure 7a; in contrast, acid gases exhibit high permeabilities thanks to their extraordinarily high S_0_ in GO membranes, which indicates that this permeation process is governed by preferential adsorption. Doping a tiny amount of PDASA into GO channels helps to promote the permeability of CO_2_ and H_2_S by 21% and 18%, respectively. Figure 7b shows the ideal selectivities of CO_2_/CH_4_, CO_2_/N_2_, H_2_S/CH_4_ and H_2_S/N_2_. Apparently, the selectivities of the above four gas pairs also increase first and then decrease with the increase of PDASA-doping amounts, and their highest selectivities can be up to 95.7, 290.3, 200.8, and 608.2, respectively. The predicted separation performance is compared with experimental results. As shown in Figure 7c,d, the separation performance for both CO_2_/CH_4_ and (CO_2_ + H_2_S)/CH_4_ of the PDASA-doped GO membranes were several orders of magnitude greater than most of the ever-reported membranes (Appendix A) and far exceed the 2008 Robeson upper bound [22], suggesting the promising potential of the adsorption-dominated separation in acid gas treatment.
(13)$$P_{i}\;=\;S_{i}D_{i}$$
(14)αij = PiPj = SiDiSjDj

## 4. Conclusions

In summary, molecular simulations are performed to investigate the adsorption and diffusion behaviors of several gases in the PDASA-doped GO membranes. Doping a tiny amount (3.0 wt%) of PDASA into GO channels effectively promotes the adsorption ability of acid gases, with the solubility coefficient of H_2_S and CO_2_ improving almost 2.5 times, while the adsorption abilities of CH_4_ and N_2_ are almost invariable with the doped PDASA. Theoretical analysis demonstrates that the isotherms of CH_4_ and N_2_ show weak adsorption, following the Langmuir model, while acid gases exhibit multilayer adsorption in GO membranes, which is relatively complex and described by the Redlich-Peterson model. Molecular insights reveal that the strong adsorption of acid gases in GO membranes is ascribed to their high isosteric heat, great binding affinity and hydrogen bonds. While their diffusion in GO channels is restrained by doping PDASA due to the narrowing of the passage. Even so, the permeability of acid gases and their ideal selectivities over CH_4_ are greatly enhanced over Robeson upper bound by doping a tiny amount of PDASA, which suggests that this removal process of acid gases is primarily dominated by preferential adsorption. From the bottom-up, this molecular understanding provides a strategy to develop high-performance GO membranes toward acid gas treatment. Such fundamental insights show the great potential of 2D membranes in the practical application of natural gas processing.

## Figures and Tables

**Figure 1 membranes-12-01155-f001:**
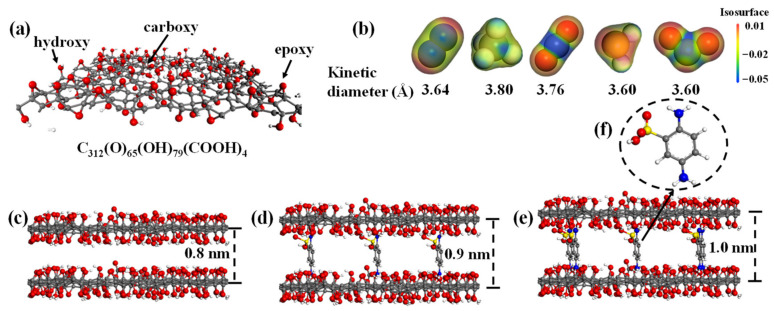
Simulation models. (**a**) GO nanosheet with the format of C_312_(O)_65_(OH)_79_(COOH)_4_. (**b**) Electrostatic potentials and kinetic diameters of gases. Configurations of GO membranes with variable PDASA-doping amounts: (**c**) 0.0%; (**d**) 4.5 wt%; (**e**) 7.5 wt%. (**f**) Molecular model of PDASA.

**Figure 2 membranes-12-01155-f002:**
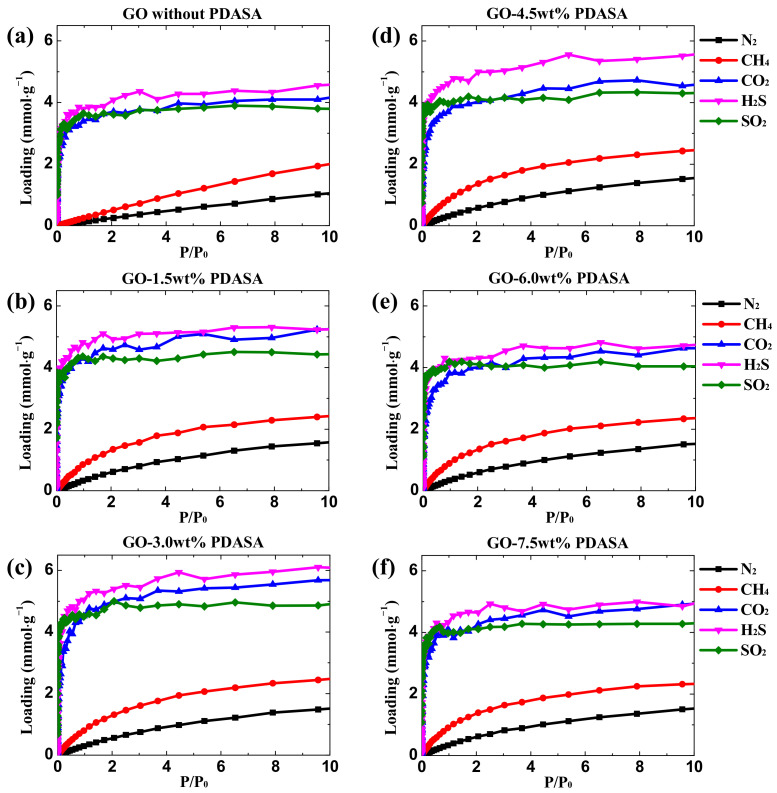
Unary isotherms of different gases in GO membranes with variable doping amounts of PDASA. (**a**) 0.0 wt%. (**b**) 1.5 wt%. (**c**) 3.0 wt%; (**d**) 4.5 wt%; (**e**) 6.0 wt%; (**f**) 7.5 wt%.

**Figure 3 membranes-12-01155-f003:**
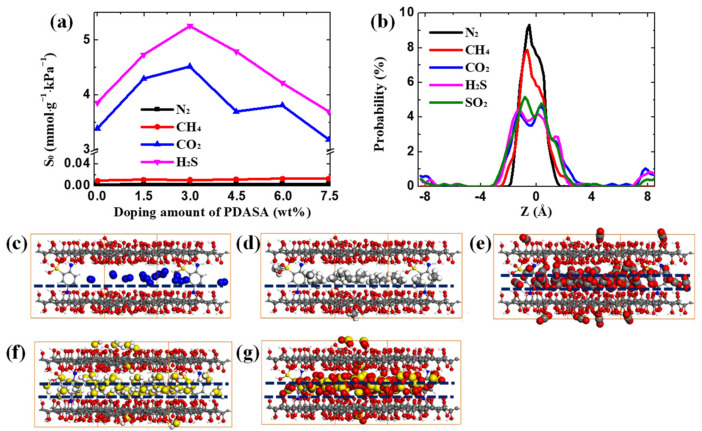
Adsorption behavior. (**a**) Solubility coefficient of different gases in GO membranes. (**b**) Distribution probability of gases in GO channels. Snapshots of variable gases adsorbed in GO channels. (**c**) N_2_; (**d**) CH_4_; (**e**) CO_2_; (**f**) H_2_S and (**g**) SO_2_.

**Figure 4 membranes-12-01155-f004:**
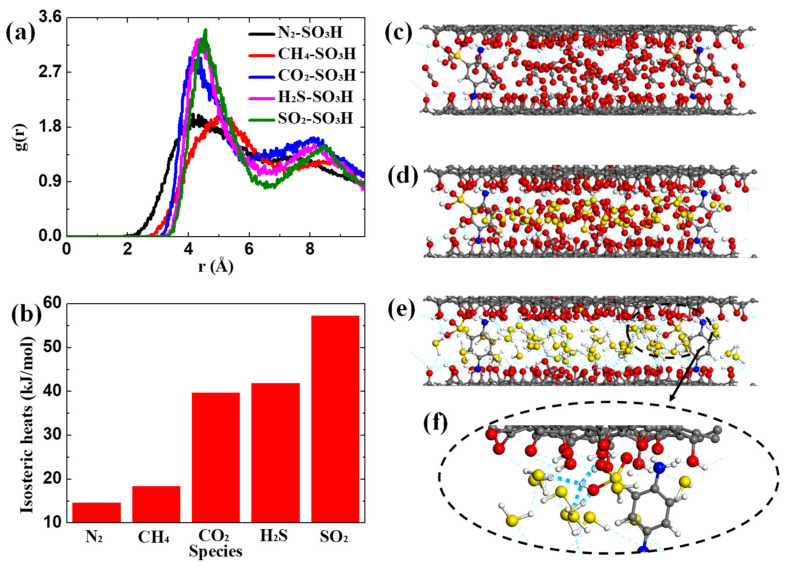
Adsorption Insights. (**a**) RDF of the doped PDASA to various gases. (**b**) Isosteric heats. Hydrogen bonds formed in the adsorption process of acid gases. (**c**) CO_2_. (**d**) SO_2_. (**e**) H_2_S. (**f**) Hydrogen bonds around PDASA.

**Figure 5 membranes-12-01155-f005:**
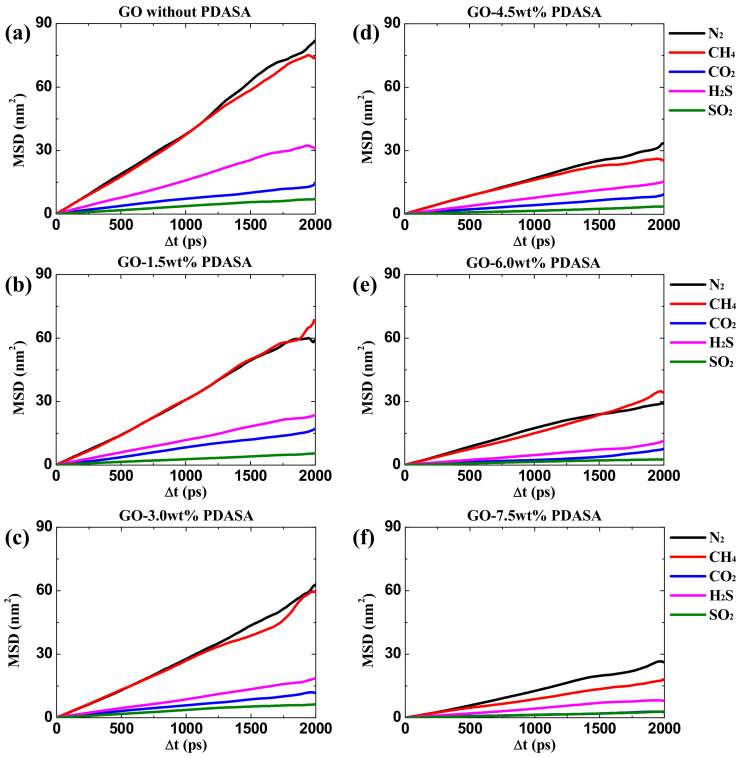
Mobility of gases in GO channels with variable doping amount of PDASA. (**a**) 0.0 wt%. (**b**) 1.5 wt%. (**c**) 3.0 wt%; (**d**) 4.5 wt%; (**e**) 6.0 wt%; (**f**) 7.5 wt%.

**Figure 6 membranes-12-01155-f006:**
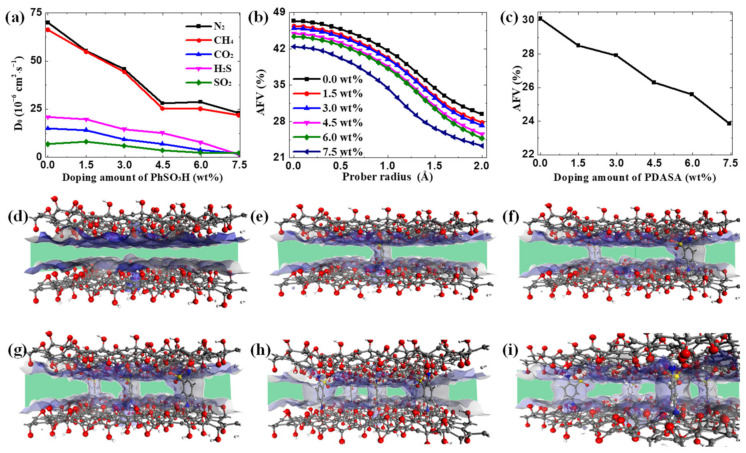
Diffusion insights. (**a**) Diffusion coefficient of different gases. (**b**) The free accessible volume of variable GO membranes. (**c**) The detected AFV with a 1.9 Å-sized prober is dependent on the doping amount of PDASA. Visualization of passage in variable GO channels. (**d**) 0.0 wt%. (**e**) 1.5 wt%. (**f**) 3.0 wt%; (**g**) 4.5 wt%; (**h**) 6.0 wt%; (**i**) 7.5 wt%.

**Figure 7 membranes-12-01155-f007:**
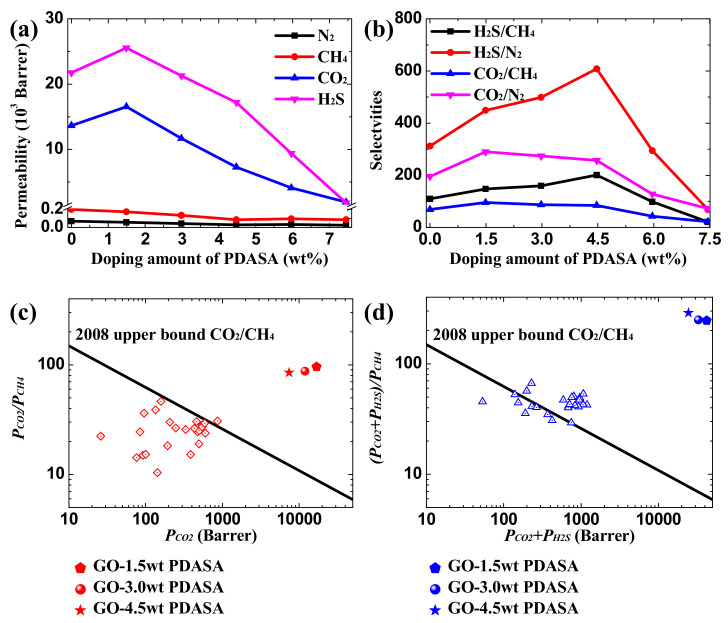
Separation performance. (**a**) Gas permeability. (**b**) Ideal selectivities of H_2_S/CH_4_, H_2_S/N_2_, CO_2_/CH_4_ and CO_2_/N_2_. Performance comparison for separations of (**c**) CO_2_/CH_4_ and (**d**) (CO_2_ + H_2_S)/CH_4_ with other potential membranes and the 2008 Robeson upper bound of CO_2_/CH_4_ (Black line).

## Data Availability

The data presented in this study are available on request from the corresponding author.

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
