# Peer review of "Functionalized GO Membranes for Efficient Separation of Acid Gases from Natural Gas: A Computational Mechanistic Understanding"

_membranes, 2022, doi:10.3390/membranes12111155_

Round 1
Reviewer 1 Report
In this manuscript, the authors have conducted molecular simulation works to elucidate acid gas removal of GO membranes functionalized by varying PDASA. The work is example of an intriguing area of research and growing field to investigate the properties of membrane materials beyond the limits of experimental techniques and to complement the experimental membrane studies by providing insights at the atomic level.
In an overall, it is a well-written paper, and the simulation results are abundant. This may help researchers to understand the effect of material designs towards acid gas removal performance at different doping amount and from molecular perspectives. The results should be useful to the community of readers of Membranes. However, I believe that the paper requires a major revision (detailed below) to clarify on several technical issues, before being accepted for the journal.
1) Please include key quantitative findings about diffusivity, solubility, permeability and selectivity from the optimal membrane design in the abstract.
2) Please check and ensure that all variables mentioned in equations are properly defined in text description. It is suggested to include an abbreviation section with units if possible.
3) It is suggested to include validation, at least for the GO membrane, to verify accuracy of the developed computational framework.
4) It seems that the authors have subjected their constructed GO membranes and gases with only geometry optimization without molecular dynamics prior to GCMC simulations. Normally, molecular dynamics are conducted to suggest that all the systems have reached equilibrium before proceeding with subsequent analysis. Any comment on this?
5) Details of the molecular dynamics in isothermal-isobaric canonical ensemble ensemble for diffusion study, e.g., number of gas molecules, settings for barostat and thermostat, operating temperature and pressure, need to be provided clearly to enable ease of reproducibility in the work.
6) Apparently, the authors have taken the entire range of MSD, which is a time series data, for the diffusivity calculations. However, near time t = 0, the particles exhibit ballistic motion, which is followed by sub-diffusive motion followed by diffusive motion. The diffusion coefficient is usually calculated by taking a slope of the MSD vs. time curve in the diffusive regime. Any comment on this?
7) The authors' motivation to benchmark acid gas performance of the designed membrane with Robeson plot is encouraging. However, it is suggested to include other commercial membranes or emerging materials with comparable operating conditions in the figure to show advancement of the membrane as compared to other works.
8) The whole manuscript should be also carefully checked to avoid editorial, language, grammar and syntax issues.
Reviewer 2 Report
1. The authors should discuss computational mechanistic in the introduction section with well-cited literature.
2. In section 3.3, the authors should support the findings with well-cited literature.
3. The heading number of section 3.5 needs to be checked; it seems it is 3.4.
4. The authors need to mention the references for equations where applicable.
5. The authors should support their findings with more discussion.
6. The authors should compare the results with well cited literature.
Round 2
Reviewer 1 Report
The authors have addressed the comments satisfactorily and the manuscript is ready to be accepted in its present form.